# Control of White Rot Caused by *Sclerotinia sclerotiorum* in Strawberry Using Arbuscular Mycorrhizae and Plant-Growth-Promoting Bacteria

**Andrea Delgado [1], Marcia Toro [1,2,*] , Miriam Memenza-Zegarra [3] and Doris Zúñiga-Dávila [1,*]**

1 Laboratorio de Ecología Microbiana y Biotecnología, Departamento de Biología, Facultad de Ciencias, Universidad Nacional Agraria La Molina, Lima 15024, Peru
2 Laboratorio de Ecología de Agroecosistemas, Instituto de Zoología y Ecología Tropical, Facultad de Ciencias, Universidad Central de Venezuela, Caracas 1040, Venezuela
3 Laboratorio de Biotecnología Microbiana, Departamento de Química Orgánica, Facultad de Química e Ingeniería Química, Universidad Nacional Mayor de San Marcos, Lima 15081, Peru
* Correspondence: mtoro@lamolina.edu.pe or marcia.toro@ucv.ve (M.T.); dzuniga@lamolina.edu.pe (D.Z.-D.); Tel.: +58-414-3056459 (M.T.); +51-975286657 (D.Z.-D.)

**Abstract:** *Sclerotinia sclerotiorum* is a phytopathogenic fungus that causes wilting and white rot in several species such as strawberry. The overuse of agrochemicals has caused environmental pollution and plant resistance to phytopathogens. Inoculation of crops with beneficial microorganisms such as arbuscular mycorrhizae (AM), plant-growth-promoting rhizobacteria (PGPR), and their metabolites is considered as an alternative to agrochemicals. *B.halotolerans* IcBac2.1 (BM) and *Bacillus* TrujBac2.32 (B), native from Peruvian soils, produce antifungal compounds and are plant-growth-promoting rhizobacteria (PGPR). *B. halotolerans* IcBac2.1 and *Bacillus* TrujBac2 with or without *G. intraradices* mycorrhizal fungi (M) are capable of controlling *S. sclerotiorum* disease in strawberries. Inoculation of mycorrhiza alone decreases disease incidence as well. Treatments with chitosan (Ch), which is used to elicit plant defense responses against fungal pathogens, were used for comparison, as well as non-inoculated plants (C). Co-inoculation of mycorrhiza and bacteria increases plant shoot and root biomass. Our results show that the inoculation of arbuscular mycorrhiza and antifungal *Bacillus* are good biocontrols of *S. sclerotiorum* in strawberry.

**Keywords:** biological control; beneficial microorganisms; microbial metabolite; *Fragaria annanasa*

## 1. Introduction

Agrochemicals are used for a fast and easy control of pests. However, they may be very toxic, can cause pest resistance, accumulation in the food chain, and soil and water contamination. Therefore, an alternative is the use of biological control [1], using predators, parasites, bacteria or fungi, or natural resources that cause less contamination, long-term results, and less risk of resistance.

Strawberries (*Fragaria annanasa* var Aromas) are popular fruits consumed in many countries [2]. In Peru, strawberry crops have increased due to favorable weather and soil conditions. Exports of frozen fruit generated profits of 5.5 million USD in 2010 with Europe and North America as the main markets. Harvest is labor-intensive and generates jobs [3]. However, annual losses caused by pathogens exceeded 200 million USD in the United States [4].

Strawberry is vulnerable to several pathogens such as *Sclerotinia sclerotiorum* fungi that attacks the plant crown causing wilt and decreasing production [5,6]. The attack of phytopathogens is generally treated with pesticides [7]. It is important to consider that foreign markets restrict the use of pesticides in fruits [3]. Thus, we considered testing biological control in strawberries [8]. Plant-growth-promoting bacteria (PGPR) produce growth substances and metabolites with phytopathogenic controlling effects and have been

successfully used in field and greenhouse assays in Peru [9,10]. In addition, arbuscular mycorrhizae (AM), a mutualistic symbiotic association present in most plants, confers on the plant a higher nutrient content, tolerance to abiotic stresses, and resistance to different diseases or pests [11–13]. Thus, AM is widely used in agriculture [14]. The objective of this work was to analyze the effect of arbuscular mycorrhizae, plant-growth-promoting bacteria, and their metabolites in the control of white rot caused by the fungus *S. sclerotiorum* in strawberry plants, comparing its effect with chitosan, an agrochemical widely used to control phytopathogens.

## 2. Materials and Methods

### 2.1. Plant and Soil Materials

Strawberry seedlings (Aromas variety), from the province of Huaral, Lima, were used. Seedlings were previously grown in the greenhouse for 40 days in a sterile mixture of rice husk, peat, and sand. Seedlings were transplanted into 800 g pots with a sterile substrate comprised of agricultural soil, peat, and sand as substrate (2:1:1). Physicochemical characteristics of the mixture used as substrate were neutral pH (7.0); slightly saline (electrical conductivity: 1.21 dS/m); high organic matter content (5.13%); low available phosphorus content (6.1 ppm); and medium available potassium content (146 ppm).

### 2.2. Inoculation of Plant-Growth-Promoting Bacteria

*Bacillus* TrujBac2.32 (B) isolated from the rhizosphere of common beans [15], was grown in 100 mL of tryptone soy broth (TSB) at 28 °C, at constant shaking (150 rpm) for 24 h. At the end of incubation, bacterial biomass was diluted with sterile TSB broth (1:1) to an optical density (at 600 nm) equivalent to $10^8$ cfu/mL.

For the production of bacterial biomass and metabolites with antifungal activity, *B. halotolerans* IcBac2.1 (BM) isolated from the rhizosphere of common beans [15] was used. The inoculum was cultured in 20 mL of TSB, incubated at 28 °C with constant shaking (150 rpm) for 24 h. Biomass obtained was added to 200 mL of minimum mineral medium (pH 7) and incubated at 28 °C, shaking constantly (150 rpm) for 72 h in order to obtain antifungal metabolites. All the fermentative broth was used, which included the biomass and all the synthesis products of the bacteria [8].

### 2.3. Glomeromycota Fungus Inoculum (M)

The fungus *Rhizophagus intraradices* was obtained from *Brachiaria decumbens* (grown in pots), in which spores and propagules were reproduced in 2 cycles of 4 months each. Afterwards, a sample of rhizospheric soil was extracted, the spores were quantified (wet sieving and decanting), and separated in sucrose gradient according to [16]. Rootlets were used to quantify mycorrhizal colonization [17] with trypan blue [18]. An inoculum was prepared with propagules richness of 2950 spores/mL and rootlets with 90% colonization. A total of 2 mL of this inoculum was added, per plant, 15 days before infecting the plant with *S. sclerotiorum* in order to achieve colonization of arbuscular mycorrhizal fungi in the roots of strawberries seedlings, previous to exposure to the pathogen.

### 2.4. Preparation of the Phytopathogen Fungus (P)

*S. sclerotiorum*, (P), isolated from common bean with symptoms of stem rot [9] was used to infect plants. It is a highly virulent pathogen with a wide host range. In pathogenicity plant tests, the characteristic symptoms of rot and mycelium proliferation were observed in the green organs of strawberry plants [5,6]. This fungus was maintained in potato dextrose agar (PDA) medium at 25 °C in the dark for 5 days. For the infection of strawberry plants, wheat seeds were used as substrate for the growth of *S. sclerotiorum*. Seeds were hydrated by placing them in a beaker with sterile water, and pre-cooked in the microwave for 5 min. After this, the water was decanted and the seeds dried under sterile conditions. Subsequently, wheat seeds were sterilized in small portions in polyethylene bags for 15 min

at 121 °C for three successive periods of 24 h. PDA discs colonized with the fungus were placed inside the bags with sterile wheat and incubated at 25 °C for 5 days.

### 2.5. Substrate Sterilization

The substrate that was used in pots to grow plants was sterilized (fluent steam) in an autoclave for 1 h, 100 °C. After 24 h, this process was repeated twice to remove Glomeromycota fungal spores and propagules from the substrate as described by [19].

### 2.6. Plant Cultures

Strawberries were grown in a greenhouse with 12 h photoperiod, maximum (32 °C) and minimum (17 °C) temperature, and 89% of relative humidity. A completely randomized design with the following inoculation treatments was performed: Control without microorganisms (C), *S. sclerotiorum* (P), arbuscular mycorrhiza (M), and the following combinations: P+M, P+B, P+BM, P+Ch, P+M+BM, and P+M+B. The experiment consisted of 9 treatments with 5 repetitions and a total of 45 pots. First, 2 plants per pot were set; bacterial inoculum and mycorrhizae were added to the seedling neck. M inoculated at sowing, B and BM 10 days after sowing. Subsequently, 15 days after installation, 4 wheat seeds covered with the mycelium of *S. sclerotiorum* were placed in the base of the plant. The appearance of the symptoms of stem wilt caused by *S. sclerotiorum* in strawberries were observed after seven days of inoculation. The disease index analysis was calculated weekly for 6 weeks, according to the scale explained below, in Table 1.

**Table 1.** Scale for measuring the degree of rot in strawberries.

| Disease Index | Percentage Damage | Plant Characteristic |
| --- | --- | --- |
| 1 | 0% | Healthy plant |
| 2 | 0–25% | Plant tissue damage |
| 3 | 26–50% | Plant tissue damage |
| 4 | 51–75% | Plant tissue damage |
| 5 | 76–100% | Plant tissue damage |
| 6 | 100% | Dead plant |

### 2.7. Disease Evaluation

Damage caused by *S. sclerotiorum* in strawberry plants was evaluated using a disease index (1–6) in the scale described in [20] that considers the largest percentage damage as 6 and healthy plants as 1.

### 2.8. Evaluation of Plant Growth Parameters

At harvest (six and a half weeks), fresh and dry weight (g) of shoot and root were recorded. Dry weight of the plants was recorded after drying the plants at 60 °C for 3 days. To evaluate the efficiency of disease control of the microbial inoculations, the following indexes were calculated: % I PSA C, (percentage of increase in aerial dry weight compared to the negative control without microorganisms, C) and % I PSA P, percentage of increase in aerial dry weight compared to the positive control (P, with *S. sclerotiorum*).

### 2.9. Quantification of Sclerotinia sclerotiorum at the End of the Experiment

A total of 10 g of soil was weighed and placed in a flask with 90 mL of 0.85% saline solution and shaken vigorously. Subsequently, with a sterile pipette, 1 mL of the suspension was taken and transferred to a test tube with 9 mL of saline solution until reaching the $10^{-4}$ dilution. Then, 1 mL aliquots of each dilution was placed in sterile Petri dishes (3 per dilution), and incorporated into PDA culture medium. A total of 30 mg/L of streptomycin was added to inhibit the bacterial population. The plates were incubated at room temperature (22 °C) for 5 days, after which the count of *S. sclerotiorum* colonies was made in the plates that contained between 8–80 colonies. The results were expressed in cfu/g of dry soil.

### 2.10. Disease Progress Curve

The area under the disease progress curve (AUDPC) evaluates the progress of the disease using the severity records, according to the following formula proposed by [21]:

$$\text{ABCPE} = \sum_{i=1}^{n-1} \left[ \frac{X_{i+1} + X_i}{2} \right] (t_{i+1} - t_i)$$

where:

$X_i$: Percentage of tissue severity due to *S. sclerotiorum* at time *I*;
$t_i$: Time elapsed in days in the evaluation *I*;
*n*: Total number of evaluations.

This method determines the severity of disease accumulated during the time of the study.

### 2.11. Statistical Analysis

A completely randomized design was performed. Least significant difference (LSD) of Fisher was applied and standard deviation of each treatment was calculated. For the analysis of aerial, root, total plant biomass, and AUDPC curve, after the ANOVA analysis, the posteriori Tukey test was applied for the comparison between averages of treatments ($p \leq 0.05$). The analyses were carried out with Statgraphics program.

## 3. Results

Antifungal capacity and growth promotion of *Bacillus* TrujBac2.32 and *B. halotolerans* IcBac2.1 were tested in previous studies carried out by [15] and their results are shown in Table 2. Both bacteria are able to control *R.solani*, *S. sclerotiorum* and *F. oxysporum* fungi. *B. halotolerans* ICBac2.1 produces antifungal lipopeptides, antibiotics, and siderophores. *Bacillus* TrujBac2.32 produces volatile metabolites and has the ability to solubilize phosphates. Both bacteria produce hydrolytic enzymes, as proteases and cellulases, and indoleacetic acid (Table 2).

Control plants without the phytopathogen have an aerial dry weight of 1.31 g, while infected plants with the pathogen (P) weigh 0.65 g. Strawberry plants inoculated with mycorrhiza (M) and *Bacillus* (B) have an increase in dry weight over 24% and 23%, respectively, compared to the infected plant. All treatments inoculated with promoting microorganisms increase the dry weight of the plant between 55 and 101% compared to (C), except the interaction of (BM) and (M) (Table 3). The dry weight of the root has a similar tendency to the shoot dry weight, except in (P) whose weight is greater (1.6 g) than C (1.46 g).

Table 4 shows the disease index of the plants evaluated for six weeks. It is observed in all cases that the damage rate increases in time. After six weeks, the plant inoculated with the fungus (P) has an index of almost 6. It should be noted that chitosan does not diminish plant damage.

The treatments that best control the disease are mycorrhizae (M), BM, and BM+M inoculations, which reduce the damage to 2.3, 4.1, and 4.3, respectively, according to the disease index (Table 1). The infection begins with the proliferation of the mycelium of the phytopathogenic fungus around the neck of the plant causing a necrosis of the plant, then the disease gradually extends to the foliage of the plant. Figure 1a shows a healthy plant, while Figure 1b shows the mycelium of *S. sclerotiorum* starting to cover the neck of the plant.

**Table 2.** Plant-growth-promoting and antifungal activities of *Bacillus* TrujBac2.32 and *B. halotolerans* IcBac2.1. The fungal growth inhibition was quantified using the percentage inhibition. Enzymatic activity: − (no activity) to + (activity). Each value represents the mean ± SE (n = 2) [15].

| Strain | Antifungal Activity by | | | | | | Antibiotics Production | AIA Production (mg/mL) | P Solubilizing Ability | Siderophore Production Halo (mm) | Hydrolytic Enzymes | | | |
| | Dual Culture Technique | | | Volatile Metabolites | | | | | | | Proteases | Cellulases | Lipases | Chitinases |
| | *R. solani* | *S. sclerotiorum* | *F. oxisporum* | *R. solani* | *S. sclerotiorum* | *F. oxisporum* | | | | | | | | |
|---|---|---|---|---|---|---|---|---|---|---|---|---|---|---|
| *Bacillus* TrujBac2.32 | 61 ± 0.01 | 65 ± 0.01 | 38 ± 0.01 | 22 ± 7.07 | 14 ± 2.12 | 0 | - | 32.6 ± 0.02 | Bi and tricalcic phosphates | 0 | + | + | - | - |
| *B. halotolerans* IcBac2.1 | 85 ± 0.01 | 71 ± 0.01 | 69 ± 0.02 | 0 | 0 | 0 | + | 8.7 | - | 20 ± 0.01 | + | + | - | - |

**Table 3.** Dry weight of shoot and root (g) of strawberry plants inoculated with *S. sclerotiorum*, growth-promoting microorganisms and their metabolites.

| Inoculation Treatments | Shoot Dry Weight (g) | Root Dry Weight (g) | Plant Total Dry Weight (g) | Percentage of Increase in Aerial Dry Weight Compared to Control C (%) | Percentage of Increase in Aerial Dry Weight Compared to P (%) |
|---|---|---|---|---|---|
| P | 0.65 ab | 1.6 ab | 2.25 ± 0.05 abc | −50.38 | 0 |
| M | 1.94 b | 1.35 b | 3.29 ± 0.66 c | 48.09 | 198.46 |
| P+M | 1.63 b | 1.30 b | 2.93 ± 0.57 c | 24.43 | 150.77 |
| P+B | 1.62 b | 1.24 ab | 2.86 ± 0.40 bc | 23.66 | 149.23 |
| P+BM | 1.01 ab | 0.56 a | 1.57 ± 0.64 ab | −22.9 | 55.38 |
| P+Ch | 0.49 ab | 0.42 ab | 0.91 ± 0.05 abc | −62.6 | −24.62 |
| P+BM+M | 0.64 a | 0.63 ab | 1.27 ± 0.27 a | −51.15 | −1.54 |
| P+B+M | 1.23 ab | 1.33 ab | 2.56 ± 1.28 abc | −6.11 | 89.23 |
| Control (C) | 1.31 ab | 1.46 ab | 2.77 ± 1.39 abc | 0 | 101.54 |

Values followed by the same letter are not significantly different according to Tukey test ($p < 0.05$). n = 5. Abbreviations: Control (without microorganisms), (P) *S. sclerotiorum*, (M) arbuscular mycorrhiza, (B) *Bacillus* TrujBac2.32 inoculation, (BM) *B. halotolerans* IcBac2.1 and its antimicrobial metabolites, and (Ch) chitosan.

**Table 4.** Disease index of wilt caused by *S. sclerotiorum* in strawberry plants inoculated with growth-promoting microorganisms and its metabolites.

| Inoculation Treatments | Weeks | | | | | |
|---|---|---|---|---|---|---|
| | 1 | 2 | 3 | 4 | 5 | 6 |
| P | 3.75 ± 2.05 | 4.25 ± 1.91 | 4.88 ± 1.36 | 5.63 ± 0.74 | 5.75 ± 0.71 | 5.75 ± 0.71 |
| M | - | - | - | - | - | - |
| P+M | 1.62 ± 0.52 | 1.75 ± 0.46 | 2 ± 0.53 | 2 ± 0.53 | 2.13 ± 0.35 | 2.38 ± 0.52 |
| P+B | 3.75 ± 2.05 | 3.88 ± 1.89 | 4.25 ± 1.67 | 4.38 ± 1.77 | 4.63 ± 1.60 | 4.63 ± 1.60 |
| P+BM | 2.38 ± 1.19 | 2.62 ± 1.41 | 3.75 ± 1.98 | 4 ± 1.51 | 4 ± 1.51 | 4.13 ± 1.64 |
| P+Ch | 3.63 ± 1.60 | 4.38 ± 1.19 | 5.38 ± 1.06 | 5.5 ± 0.93 | 5.5 ± 0.93 | 5.75 ± 0.71 |
| P+BM+M | 3.38 ± 2.20 | 3.50 ± 2.14 | 4 ± 1.93 | 4.13 ± 1.89 | 4.25 ± 1.98 | 4.38 ± 2.00 |
| Control (C) | - | - | - | - | - | - |

The values were compared using least significance difference (LSD) test of Fisher, and show means ± standard deviation per treatment. n = 5. Abbreviations: Control (without microorganisms), (P) *S. sclerotiorum*, (M) arbuscular mycorrhiza, (B) *Bacillus* TrujBac2.32 inoculation, (BM) *B. halotolerans* IcBac2.1 and its antimicrobial metabolites, and (Ch) chitosan. Treatments M and Control show no symptoms since they are not inoculated with the phytopathogenic fungus.

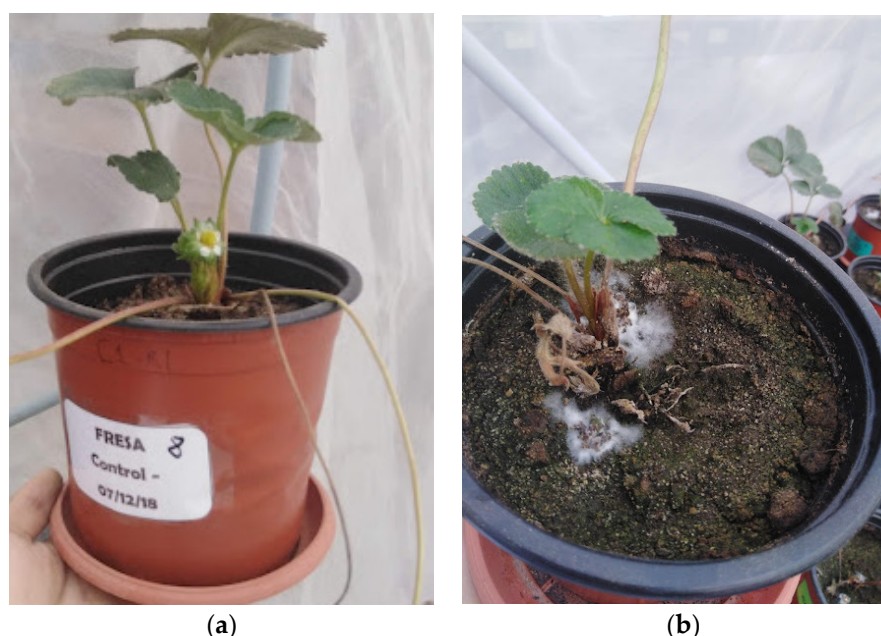

(**a**)　　　　　　　　　　　　　　　　　　　　　　　　(**b**)

**Figure 1.** (**a**) Healthy plant without disease symptoms. (**b**) Appearance of the mycelium of the *S. sclerotiorum* developing in the neck of strawberry.

Table 5 shows that plants treated with BM+M have a higher percentage of AM root colonization (20.4%), followed by B+MI (12%). The bacteria significantly stimulates mycorrhizal roots colonization of the plant, compared to that plant that is only inoculated with the mycorrhizal fungi, while the pathogen diminishes mycorrhizal colonization.

**Table 5.** Arbuscular mycorrhizal colonization (%) in strawberry plants inoculated with *S. sclerotiorum*, growth-promoting bacteria and their metabolites.

| Inoculation Treatments | % AM Root Colonization |
|---|---|
| P+BM+M | 20.46 ± 1.65 c |
| P+M | 8.94 ± 2.06 a |
| P+B+ M | 14.96 ± 0.95 b |
| M | 17.49 ± 3.39 bc |

Values followed by the same letter are not significantly different, according to Tukey test ($p < 0.05$). n = 5.

At harvest, the population of *S. sclerotiorum* that was in the soil around the crown of the plant was quantified. Table 6 shows that there were $80 \times 10^3$ cfu of *S. sclerotiorum*/g soil. All treatments reduce the population of the phytopathogenic fungus, with (BM) and (B+M) ($50 \times 10^3$ cfu/ g dry soil) being the most effective treatments. Chitosan has no effect.

**Table 6.** Number of colonies of the phytopathogen *S. sclerotiorum* in rhizospheric soil (cfu/g dry soil) after the harvest of strawberries inoculated with growth-promoting microorganisms and its metabolites.

| Inoculation with Beneficial Microorganisms in Addition to *S. sclerotiorum* | $10^4$ cfu *S. sclerotiorum*/g Dry Soil |
|---|---|
| P | $80 \pm 35$ |
| M | $60 \pm 30$ |
| B | $60 \pm 32$ |
| BM | $50 \pm 25$ |
| Ch | $80 \pm 15$ |
| BM+M | $60 \pm 35$ |
| B+M | $50 \pm 35$ |

The values were compared using least significance difference (LSD) test of Fisher, and show means $\pm$ standard deviation per treatment. n = 3. Abbreviations: (P) *S. sclerotiorum*, (M) arbuscular mycorrhiza, (B) *Bacillus* TrujBac2.32, (BM) *B. halotolerans* IcBac2.1 and its antimicrobial metabolites, and (Ch) chitosan.

Figure 2 shows the number of colonies of *S. sclerotiorum* grown in PDA medium isolated from soil in pots after plant harvest.

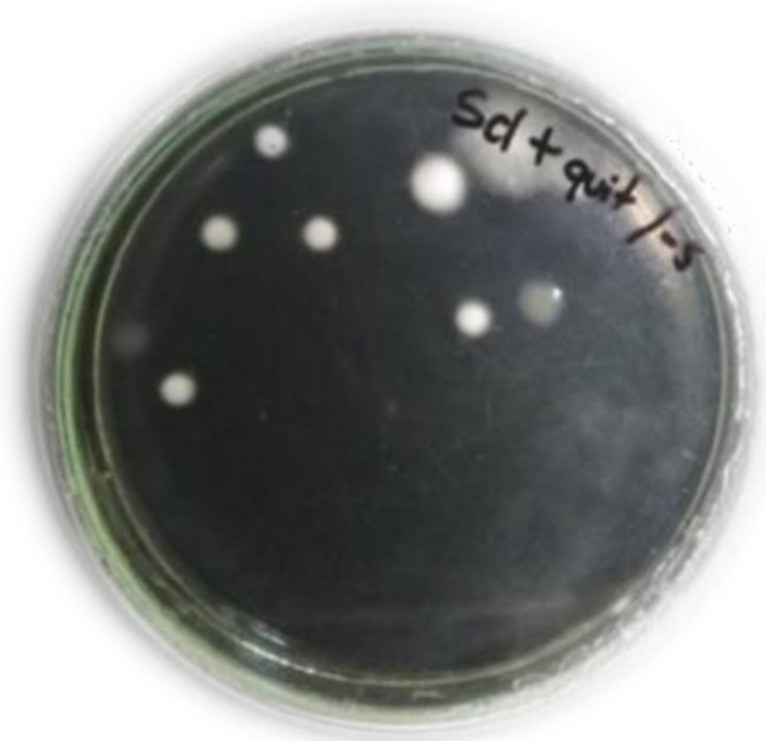

**Figure 2.** Colonies of *S. sclerotiorum* in PDA medium.

In relation to AUDPC (Figure 3), the values of each bar correspond to the average of eight repetitions/treatment, representing the progress of the disease on plants. According to the ANOVA analysis, P+M is the treatment with the lowest ABCE value (31), however, it is statistically similar to P+B, P+BM, and P-BM+M. P is statistically to similar P+Ch, indicating that chitosan does not diminish the disease. The analysis indicates that beneficial microorganisms alone (M, B, or BM) or in combination, BM+M, control *S. sclerotiorum*.

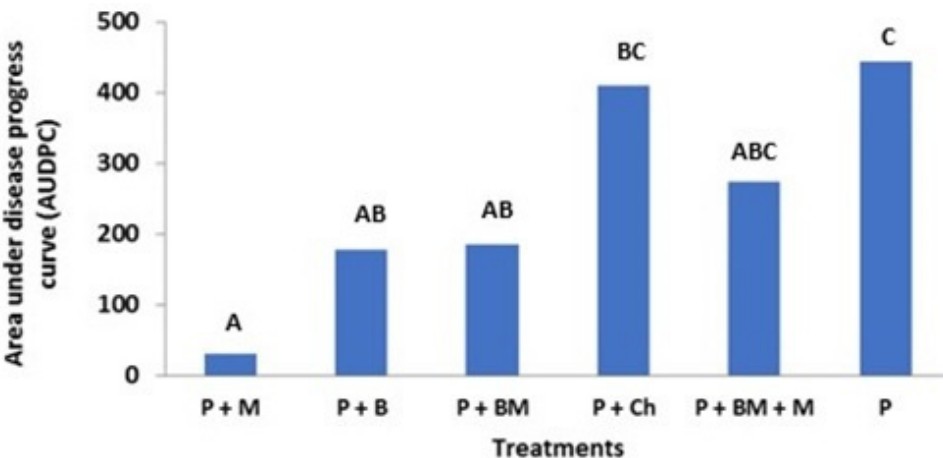

**Figure 3.** Area under disease progress curve (AUDPC) of the disease caused by *S. sclerotiorum* in strawberry plants inoculated with arbuscular mycorrhiza, growth-promoting bacteria and their metabolites. Abbreviations: (P) *S. sclerotiorum*, (M) arbuscular mycorrhiza, (B) *Bacillus* TrujBac2.32, (BM) *B. halotolerans* IcBac2.1and its antimicrobial metabolites, and (Ch) chitosan. Different letters over the bars indicate significant differences between treatments (Tukey $p \leq 0.05$). n = 8.

## 4. Discussion

*Bacillus* are commonly used to control phytopathogens [22,23]. Different mechanisms have been proposed to explain the ability to inhibit mycelial growth of fungal pathogens, such as the production of antimicrobial metabolites, competition for nutrients or space, or a combination of them [24]. Ref. [25] found that *B. amyloliquefaciens* protected tomato, squash, and eggplant from *S. sclerotiorum*. The protection is related to the antibiosis capacity of *B. amyloliquefaciens*. This bacterium also controls other fungi through the production of a-1,3-glucanase. In addition, due to its capacity to produce enzymes such as chitinases, proteases, amylases, and celluloses, *B. amyloliquefaciens* and other *Bacillus* species cause leakage of fungal cell contents when hydrolyzing cell walls of the phytopathogen. Here, we found that both *Bacillus* strains produce hydrolyzing enzymes. Ref. [26] also observed that cell suspensions, cell-free culture filtrates, and broth cultures of *B. subtilis* control *S. sclerotiorum* in soybean, through the reduction in its mycelial growth (by 50 to 75%) and suppressing sclerotial formation by > 90%. Antibiotic peptides present in its cell-free supernatant may be responsible for this effect [10,15]. Antimicrobial metabolites produced by bacteria include iron-chelating siderophores, antibiotics, volatile biocides, and non-volatile antimicrobial compounds such as cyclic lipopeptides (CLPs) such as iturins, fengycins, and surfactins involved in the inhibition of mycelial growth of phytopathogens [27,28].

Here, we found that BM reduces the infection rate of the phytopathogenic fungus to 50%, while the disease index of P is greater than 81%. In our group we reported previously a reduction in the incidence of *S. sclerotiorum* wilt in beans by 96% due to the application of the strain *B. halotolerans* IcBac2.1 (BM) [9]. There, the bacterium was inoculated at different stages of growth, seed, flowering, formation, and filling of pods. Notably, the same strain was able to reduce the population of the phytopathogen in the soil, similar to our results with strawberry.

Bacteria may promote plant growth by phytohormones and solubilize phosphate [29]. Several species of *Bacillus* produce indole-3-acetic acid (IAA), which increases root length and the number of secondary roots [30]. Ref. [31] demonstrates that *B. velezensis* and *B. amyloliquefaciens* can produce phytohormones, fix nitrogen, and solubilize phosphate, in addition to reducing disease against phytopathogens such as *Helicobasidium purpureum*, *F. oxysporum*, and *Rhizoctonia solani* on pepper. Both bacteria used in this work have the capacity of producing AIA; *Bacillus* TrujBac2.32 (B) produces large amounts of AIA (32.6 μg mL$^{-1}$) [15,32]. In this work, this strain was able to improve shoot dry weight of the plant by 149% (Table 2).

Chitosan is considered an environment-friendly alternative to the use of chemicals for controlling phytopathogens such as *Sclerotinia*, among others, and post-harvest diseases in fruits [33]. In our case, chitosan is not able to diminish disease or the fungal population in the soil. Results obtained in different studies with chitosan are variable and depend on several factors, such as the concentration used, the application method, and the experimental temperature of the greenhouse, among others [34].

Regarding arbuscular mycorrhizae, the inoculation of the fungus *R. intraradices* (M) is able to significantly reduce the disease index by effectively controlling infection. In this treatment the highest shoot dry weight of the plant increased by 150% (Table 2). However, the percentage of AM root colonization is the lowest (Table 3), which could indicate that even with 8% root colonization, the promoter effect exerted by *R. intraradices* is sufficient to achieve such biomass. Similar to our results, [35], mycorrhiza achieves adequate control of the disease caused by *Fusarium* sp. in maize, with a low number of Glomeromycota spores or root colonization, which could also be associated with the effectiveness and affinity of the AM fungus with the host in question. Arbuscular mycorrhizae can induce systemic resistance in the plant to soil and/or aerial pathogens [36]. The production of jasmonic and/or salicylic acids, among others, as phytopathogenic resistance mechanisms are stimulated in the plants by mycorrhizal inoculation [37]. In mycorrhized plants, the expression of different genes involved in plant defense against foliar and root pathogen attack may play a role locally and systemically. There are mechanisms in which acid endochitinase PR4 and β1, 3-endoglycanase EG488 may also be involved [38]. Moreover, AM fungi can improve the resistance of host plants to phytopathogens by changing the anatomical structure of plant roots and producing substances involved in plant defense against pathogenic attack [39]. Some authors [40] described the fact that mycorrhizal association causes a chemical balance in the plant that inhibits the growth and reproduction of phytopathogenic fungi. Ref. [41] observes that pre-inoculation with the AMF *Glomus mosseae* prevents the decrease in antioxidant compounds, along with the decrease in SOD activity and ascorbate content in strawberry roots after inoculation with *F. oxysporum* f. sp. fragariae. These mechanisms, although not evaluated in our case, could be occurring for the control of *S. sclerotiorum*.

In cotton, inoculation with PGPR stimulates root AM colonization [42], which is in agreement with the results presented here where the strains of *Bacillus* TrujBac2.32 (B) and *B. halotolerans* IcBac2.1 and its metabolites (BM) increase root mycorrhizal colonization significantly, 12 and 20.4 %, respectively (Table 3). The interaction between PGPR and AM can favor the plant both by promoting mycorrhiza activity, which, in turn, favors the development of plants (mycorrhiza helper bacteria) [43,44], and by contributing to the control of phytopathogens [36].

BM+M is the most effective treatment in the control of the disease and better mycorrhizal colonization, which could, in turn, influence the decrease in *Sclerotinia* spores in the rhizosphere of the plant. Although this treatment does not produce more dry biomass of the plant, it is effective in the parameters that indicate control of the phytopathogen. The antifungal effects of *B. halotolerans* ICBac2.1 may be due to the production of lipopeptides [15] and could explain our results.

Similarly, other authors found that the interaction of AMF and PGPR controlled plant diseases [27]. AM and PGPR favor the tolerance of strawberry plants to wilt induced by *P. capsici*. [39], indicating that the combined treatment of AMF and PGPR as *Bacillus* exerts a positive control on plant root diseases. The mixed inoculation of *G. mosseae* and *Bacillus subtilis* not only reduces the severity of *Fusarium*-caused disease by 85.0–93.4%, but also improves plant nutrient content, total soluble sugar, total soluble protein, and total free amino acid content [45,46]. Our results show that the joint use of beneficial microorganisms, AMF and PGPR, can control plant diseases, and constitute an alternative for sustainable agriculture, contributing to the reduction in the application of agrochemicals. Ref. [47] proposes that the use of biological control, together with crop rotation and the use of resistant cultivars, will allow for effective sustainable management of agriculture.

## 5. Conclusions

The interaction between PGPR and AM favors the development of plants and contributes to the control of phytopathogens, such is the case of BM +M, which is the most effective treatment in controlling the disease caused by *S. sclerotiorum* and promotes mycorrhiza activity. The presence of beneficial microorganisms in the rhizosphere with influence on the state of health of the plant is complex and manifests itself in our results, so it could be considered that the microorganisms used influence the defense responses of strawberry plants to *S. sclerotiorum*. Our results constitute a first approach to the joint use of native strains of both PGPR and AMF for the control of the disease caused by *S. sclerotiorum* in strawberries plantations in Peru. We suggest further research of these studies.

**Author Contributions:** Conceptualization, M.T. and D.Z.-D.; methodology, M.T., M.M.-Z. and D.Z.-D.; validation, A.D. and M.M.-Z.; formal analysis, M.T. and D.Z.-D.; investigation, A.D. and M.M.-Z.; resources, D.Z.-D.; writing—original draft preparation, M.T. and D.Z.-D.; writing—review and editing, M.T.; supervision, M.T., M.M.-Z. and D.Z.-D.; project administration, D.Z.-D.; funding acquisition, D.Z.-D. All authors have read and agreed to the published version of the manuscript.

**Funding:** This work was supported by grants 009-2016 Fondecyt from Fondo Nacional de Desarrollo Científico, Tecnológico y de Innovación Tecnológica and Innovate Perú Proyect 158-PNCIP-PIAP-2015 from Programa Nacional de Innovación para la Competitividad y Productividad del Ministerio de la Producción.

**Acknowledgments:** The authors express their gratitude to Wilson Castañeda for the preparation of the Glomeromycota inoculum used in the experiment and appreciate the collaboration of Auxi Toro García and Esperanza Martínez-Romero (Centro de Ciencias Genómicas, Cuernavaca, Mexico) in their support as linguistic advisors.

**Conflicts of Interest:** The authors declare no conflict of interest.

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
