# Peer review of "Control of White Rot Caused by Sclerotinia sclerotiorum in Strawberry Using Arbuscular Mycorrhizae and Plant-Growth-Promoting Bacteria"

_sustainability, doi:10.3390/su15042901_

Round 1

Reviewer 1 Report

see attach file 

Reviewer 2 Report

In the present study, the authors reported the effect of co-inoculation of arbuscular mycorrhizae, plant growth promoting bacteria and their metabolites in the control of white rot caused by Sclerotinia sclerotiorum in strawberry. The results were very simple, but the effects of different treatments on the agricultural characters of strawberry. The method lacked the description of the repetitions in each treatment. The statistical analysis of the results was poor. The author should further analyze the reasons why M and B can increase the r favored plant shoot and root biomass of strawberry, such as soil enzyme activity, root activity and MDA content of root, TEM photographs of roots, etc.

1. The results in the Abstract were too simple (only two sentences) and should be further refined.

2. Line 24: “ Bacillus TrujBac”,“Bacillus” should be italic., and also Line 81, 161, et al.

3. The results in tables should be expressed as mean ± SD (n =?). How many repetitions in each treatment? Based on statistical analysis in table 2, the number of repetitions in each treatment seems insufficient.

Reviewer 3 Report

The article prepared by Delgado et al. is of great scientific interests but it requires substantial improvement for publication. Please carry on an extensive editing of English language and style which is required for clarification and making understandable. In addition, some comments are as below:

1. Title: Please modify the title to make short and attractive.

2. Abstract: Please improve the abstract and rewrite the line 14 to 17 for clarification.

3. If possible, include some more slides in Figure 1. 

4. Please rewrite the figure legend as stated in the author instruction guideline or following published article in sustainability journal.  

5. I would like to suggest inclusion of a "conclusion" section.

6. The scientific names including genus should be italic. Please check throughout the manuscript.

7. Please include some more recent references form 2022, and 2021 published on this topic.

Round 2

Reviewer 1 Report

authors improve the article but the article need English editing before could be accept 

Author Response

The manuscript was revised following the suggestions of the reviewers. Researchers with experience were requested to review the English language. We hope to meet the requirements of the reviewers in this new version.

Reviewer 2 Report

The revised manuscript adds some experimental details, such as he number of repetitions. However, it has not improved in some important aspects. The main part of the abstract was the research background and methods. The methods and results were very simple. Further experiments focusing on the promoting mechanisms of the PGPRs were still missing, such as oil enzyme activity, root activity and MDA content of root, TEM photographs of roots, etc. 

Author Response

(The authors gave the same response as above.)
